# Cell-Free Nucleic Acids and their Emerging Role in the Pathogenesis and Clinical Management of Inflammatory Bowel Disease

**DOI:** 10.3390/ijms20153662

**Published:** 2019-07-26

**Authors:** Zuzana Kubiritova, Jan Radvanszky, Roman Gardlik

**Affiliations:** 1Institute for Clinical and Translational Research, Biomedical Research Center, Slovak Academy of Sciences, Dubravska cesta 9, 84505 Bratislava, Slovakia; 2Department of Molecular Biology, Faculty of Natural Sciences, Comenius University, Ilkovicova 6, 84215 Bratislava, Slovakia; 3Institute of Molecular Biomedicine, Faculty of Medicine, Comenius University, 81372 Bratislava, Slovakia

**Keywords:** cell-free nucleic acids, circulating nucleic acids, cell-free DNAs, cell-free RNAs, exosomes, inflammatory bowel disease, neutrophil extracellular traps, NETosis

## Abstract

Cell-free nucleic acids (cfNAs) are defined as any nucleic acids that are present outside the cell. They represent valuable biomarkers in various diagnostic protocols such as prenatal diagnostics, the detection of cancer, and cardiovascular or autoimmune diseases. However, in the current literature, little is known about their implication in inflammatory bowel disease (IBD). IBD is a group of multifactorial, autoimmune, and debilitating diseases with increasing incidence worldwide. Despite extensive research, their etiology and exact pathogenesis is still unclear. Since cfNAs were observed in other autoimmune diseases and appear to be relevant in inflammatory processes, their role in the pathogenesis of IBD has also been suggested. This review provides a summary of knowledge from the available literature about cfDNA and cfRNA and the structures involving them such as exosomes and neutrophil extracellular traps and their association with IBD. Current studies showed the promise of cfNAs in the management of IBD not only as biomarkers distinguishing patients from healthy people and differentiating active from inactive disease state, but also as a potential therapeutic target. However, the detailed biological characteristics of cfNAs need to be fully elucidated in future experimental and clinical studies.

## 1. Introduction

Inflammatory bowel disease (IBD), which involves Crohn’s disease (CD) and ulcerative colitis (UC), is a group of multifactorial disorders characterized by chronic inflammation affecting the gastrointestinal tract in variable extent, typically leading to multiple symptoms such as weight loss, abdominal pain, recurrent diarrhea, and bleeding. Despite extensive research in this field, the exact pathogenesis of IBD is still unknown. According to current knowledge, IBD develops due to deregulated complex interactions between genetic, environmental, and immunological factors, including the complex interactions between the gastrointestinal microbiota and the host organism [1,2]. The clinical diagnosis of IBDs is based on numerous investigations, including ultrasonography, endoscopy, as well as laboratory analyses such as biochemical blood and stool markers, or histological tests of affected tissues. The diagnostic procedures can be time consuming and uncomfortable for patients, especially for the continuous monitoring of their health status, disease progression, or therapeutic effectiveness. Moreover, therapeutic possibilities are limited to the suppression of acute inflammation and maintaining remission, while a marked interindividual variability of effectiveness was reported for each line of actually available therapeutics [3,4]. Therefore, current research related to IBD is strongly focused on various aspects of the disease with the aim to elucidate the exact mechanisms of the pathogenesis, as well as to identify either novel biomarkers (for both disease status evaluations, prognostics, and therapeutic response predictions and monitoring) or new therapeutic targets and agents. One of the highly progressing fields that has an impact in each of these aspects is genomics and nucleic acids research in general. Besides the direct analyses of genomic material obtained from cells, the possibility of identification and analysis of cell-free nucleic acids (cfNAs) in the circulation, or in other body fluids, is now attracting particular attention in several biomedical fields. In a broader context of complex care, the analysis of nuclear genomic material offers the identification of yet asymptomatic individuals having high risk for certain types of IBD [5], while the identification of cfNAs seems to represent promising non-invasive biomarkers to distinguish patients having active disease from those in inactive phase or from healthy people, allowing to track disease onset, progression, and remission following therapy. However, still little is known about the specific cfNAs and their exact roles and associations in the pathogenesis of IBD. Therefore, the aim of this review is to collect and integrate available information about these important aspects.

## 2. Cell-Free Nucleic Acids and their General Recognition and Use

Our knowledge about the existence of cfNAs is not new. Since their first description in 1948 [6], their presence was observed in various biological fluids, such as blood, urine, saliva, but also in stool. Their biological significance and usability as biomarkers of health status is given by several factors. The two main advantages are their differential presence in healthy individuals versus patients having certain diseases, or specific physiological conditions, as well as their convenient accessibility from liquid biopsy, saliva, or stool sampling. Moreover, their differential patterns of methylation status or fragmentation may serve as information about the organism, organ, tissue, or mechanism of origin of particular fractions of cfNAs [7,8,9,10].

Although not the first, probably the most popular and most rapidly adopted use of cfNAs globally is the so-called non-invasive prenatal testing (NIPT) [11]. This method opened a completely new era of prenatal care, especially following its housing with advanced DNA analytical technologies, such as massively parallel genome-scale DNA sequencing. However, prenatal testing-based analyses of cfNAs started to show also other utilities, even going beyond conventional prenatal testing. The spread of cfNAs analyses already took place also into unrelated clinical fields, such as oncology, in a form of liquid biopsy-based non-invasive cancer detection (NICD), since cfDNA levels are typically elevated in cancer patients [12]. The possible relevance of cfNAs has been previously reported also for a range of other diseases and pathological conditions, such as trauma, myocardial infarction, stroke, transplantation, diabetes, sickle cell disease, sepsis, and aseptic inflammation [13]. Moreover, in addition to cancer and inflammation, other aging-related degenerative processes, such as cellular senescence, were also suggested to be associated with age-related increasing levels of cfNAs (specifically cfDNA), possibly through DNA damage response-induced higher genome instability [14].

## 3. Origin and Basic Types of Cell-Free Nucleic Acids

There are plenty of different types of cfNAs molecules that can both originate from various biological processes and participate on various physiological and pathophysiological processes in a complex yet poorly understood manner. According to their origin, cfNAs can be divided into various categories. At first, it is important to differentiate between endogenous and exogenous cfNAs. While endogenous cfNAs originate from tissues and the cells of the organism of interest itself, exogenous cfNAs may typically come from the host microbiome, from different infections and parasites, as well as from the ingested food of the host organism [15]. On the border between these are cfNAs originating from developing fetuses in maternal organisms [16] which, although having endogenous origin, are coming from a different individual of the same biological species. Moreover, in humans, the transfusion origin of cfNAs should also be taken into account [17].

When considering the type of the macromolecule of interest, cfNAs are generally divided into DNAs or RNAs. Endogenous cfDNA includes nuclear (genomic) DNA (cf-ncDNA) as well as mitochondrial DNA (cf-mtDNA), whereas exogenous DNA is usually represented by microbial and specifically bacterial DNA. It is hard to universally describe the length range and exact composition of cfDNAs, since their extremely dynamic nature and various extraction methods result in variable recovery rates. However, they were described in the range from ultrashort, <100-bp fragments, up to several thousands of bps, both for nuclear as well as for mitochondrial DNA [12,18]. Nuclear cfDNA fragments have predominancy under physiological conditions, and are highly fragmented molecules to the size of approximately 166 bp, which is approximately the length of a segment of DNA wound around a protective histone octamer. They mainly come from the apoptosis of hematopoietic cells [19]; however, they can also be released from necrotic cells—but only through phagocytosis [20]. Another process of the formation of circulating cfDNA is NETosis (name comes from neutrophil extracellular traps) [21], but the active release of newly synthesized DNA via vesicles and lipoproteonucleotide complexes was also reported [22]. In a comparison to cf-ncDNA, cf-mtDNA is much smaller. Its predominant size ranges approximately from 30 bp to 80 bp, because mtDNA is, unlike ncDNA, a small molecule that is not protected by histones. Similar to cf-ncDNA, cf-mtDNA can be transported by extracellular membrane vesicles (EMVs) [12]. Similar to cfDNAs, cfRNAs can also be divided into many other types. According to the functionality, we differentiate coding RNAs (mRNAs) and non-coding RNAs. Non-coding RNAs are further divided into many subtypes, including lncRNAs (long non-coding), miRNAs (micro), tRNAs (transfer), YRNAs, piRNAs (PIWI-interacting), circRNAs (circular), other small non-coding RNAs (ribosomal, small nuclear, small nucleolar); however, these are present in amounts less than 1% [12]. Among cfRNAs, the cf-mRNAs are fragmented and less abundant; therefore, many studies focus on the analysis of small non-coding RNAs, which are more stable and therefore also more abundant. As cfRNAs are relatively unstable molecules that are susceptible to degradation by ribonucleases, they can generally be found encapsulated within EMVs (apoptotic bodies, exosomes, microvesicles) or form ribonucleoprotein complexes (by binding to high-density lipoprotein or RNA-binding proteins) [23,24].

Based on some of the above-mentioned properties, both endogenous and exogenous cfNAs in circulation can be further divided into free cfNA fragments (naked sequences having no specific vesicles), vesicle-bound cfNA fragments (nucleic acids in EMVs divided into exosomes, microvesicles, and apoptotic bodies) and cfNA macromolecular complexes (e.g., nucleosomes, virtosomes, and neutrophil extracellular traps), all of which have specific origin and routes of transport (Figure 1) [12].

## 4. Exosomes in IBD

Exosomes are endosomal-derived nanovesicles that are secreted from many types of cells in both physiological and pathological conditions. They are not produced by “simple” budding from the plasma membrane, as microvesicles are; rather, their highly regulated biogenesis is connected to the endolysosomal pathway. This starts with an endocytosis-based generation of an early-endosome that matures to a late-endosome, a multivesicular body, through endosomal membrane budding. By exocytosis, such multivesicular bodies can fuse with the plasma membrane, releasing their individual vesicles (exosomes) from the cell. Alternatively, multivesicular bodies can fuse with lysosomes to become degraded, and the function of their cargos can be lost [25,26]. Since these exosomes can transport various molecules long distances inside the body, such as proteins, lipids, and nucleic acids (mainly miRNAs and mRNAs), they are generally involved in various biological activities, such as cell–cell communication and cell–environment interactions. These include the modulation of immune responses, intestinal barrier functions, and regulation of the intestinal microbiome. The specific functions of exosomes in these complex interactions, not only in IBD, depend primarily on their functional components and exosome structure [27]. While the transferred proteins can directly activate or inhibit biological pathways [28], transferred regulatory NAs can modulate pathways through the specific regulation of gene expression in target cells [29]. Exosomes have been previously isolated from plasma, colonic luminal fluid aspirates, intestinal epithelial cells, and saliva [30,31,32,33]. As they are released by different tissues and can be collected from several body fluids and biological materials, the information carried by their specific cargos may represent potential disease biomarkers, while they themselves may represent possible therapeutic targets or therapeutic agents [34].

Several studies focus on the analysis of protein content of exosomes from IBD patients and healthy controls or mouse models of colitis. When comparing their cargos, exosomes from the saliva of IBD patients, for example, revealed several proteins presenting exclusively in just one of the studied groups, i.e., only in UC patients, CD patients, or healthy controls. From these, for example, proteasome subunit alpha type 7 (PSMA7) was selected to be a promising biomarker, since this exosomal protein is related to proteasome activity and inflammatory response [33]. Tens of differentially expressed proteins in serum exosomes were also found in mice with acute colitis when compared to exosomes isolated from control mice. The majority of them were involved in the complement and coagulation cascade, which has been implicated in macrophage activation, pointing thus to specific roles in the pathogenesis of IBD [35]. Moreover, alterations of intestinal exosome proteomes were associated also with aberrant host–microbiota interactions in IBD. It was demonstrated that exosomes released from intestinal epithelial cells are involved in activating host innate immune responses and in subverting the intracellular replication control of adherent-invasive *Escherichia coli*. It is known that these strains are in a high prevalence in the intestinal mucosa of patients with CD, and enhance intestinal epithelial permeability by modulating and/or disorganizing cell junction proteins [36,37]. Along with altered protein profiles, distinct mRNA profiles were also found in exosomes shed from sites of inflammation in patients with IBD. These were shown to have pro-inflammatory effects on the colonic epithelium in vitro, which is mainly due to an increase of interleukin-8 (IL-8) [31].

The above-mentioned associations point toward an increased involvement of exosomes in pro-inflammatory cascade in IBD pathogenesis. However, exosomes from normal intestine were found to have immunosuppressive effects with a potential to prevent the development of IBD in a murine model of colitis, for example by inducing T-regulatory and dendritic cells. On the other hand, their inhibition can exacerbate murine IBD [30]. Besides exosomes from intestinal epithelial cells, exosomes secreted by mesenchymal stem cells derived from a human umbilical cord were shown to have a repairing effect on inflamed intestinal tissue. It was proven that they have profound effects on alleviating a dextran sulfate sodium (DSS)-induced colitis through the modulation of IL-7 expression in macrophages, or by regulating the ubiquitin modification level [38,39]. In the serum of patients with active IBD, elevated levels of secreted annexin A1-(ANXA1-) containing extracellular exosomes were detected, most likely as a result of systemic distribution in response to the inflammatory process. Endogenous ANXA1 is released as a component of extracellular exosomes derived from intestinal epithelial cells, while these ANXA1-containing exosomes have the potential to activate wound repair circuits [32].

Taken together, these findings suggest that exosomes, through the specific constellation of their cargos, may exert both pro-inflammatory as well as anti-inflammatory effects on tissues, including intestinal tissues. Since exosomes with pro-inflammatory contents seems to be enriched during active IBD (or experimentally induced colitis), while those with anti-inflammatory cargos are dominant in healthy patients (or control mice), their precise balance in different physiological situations likely plays a crucial role in inducing, maintaining, or regulating the required functions of intestinal tissues. Therefore, exosomes or their specific cargos can be considered when looking for biomarkers of intestinal mucosal inflammation, but also when looking for potential therapeutic strategies in situations of chronic mucosal injury. Moreover, in the latter case, their involvement can be considered both as possible targets for inhibition (in case of pro-inflammatory contents), or as possible therapeutic agents (in case of anti-inflammatory contents), to induce and/or maintain intestinal homeostasis. The biological significance and clinical potential of exosomes as markers and therapeutic tools in IBD has been recently reviewed [26,27,40]. However, knowledge of the role of exosome-specific NAs has been generally limited to microRNAs as the major human plasma-derived exosomal RNA species. The role of miRNAs is discussed later in this text.

## 5. Neutrophil Exracellular Traps in IBD

Neutrophil extracellular traps (NETs) are complex structures released from neutrophils due to chromatin decondensation and spreading. These so-called “traps” are being released into the environment, consisting of a tangle of released nucleic acids, histones, and proteases [41]. Then, these traps are able to capture the bacteria and kill them. Despite being macromolecular complexes, NETs, or their degradation products, were found to contribute to the total pool of cfDNAs [21]. They were described by Takei et al. in 1996 as a pathway of cellular death that is different from apoptosis and necrosis [42], and in 2004, Brinkmann et al. described a process named NETosis, which is one of the mechanisms that neutrophils undertake for host defense [43]. There are several known models of NETosis, which are either dependent or independent of reactive oxygen species (ROS), and also they differ in releasing either nuclear or mitochondrial DNA. Neutrophils and NETs protect hosts from infectious diseases, while the aberrant formation and/or clearance of NETs may have pro-inflammatory characteristics and may be implicated in many infectious and non-infectious diseases, such as cancer, cardiologic, metabolic, inflammatory, and lung diseases [44]. The nuclear material released from NETs could be more immunogenic than the apoptotic one. Both native and oxidized endogenous DNA bound to NETs activate dendritic cells to synthetize interferon-α in a TLR (toll-like receptor)-dependent manner. NETs also increase the T-cell response to antigens and activate B cells to induce immunoglobulin (Ig) class switching and antibody production. Oxidized DNAs are also more resistant to degradation, contributing thus to sustain a dysregulated immune response, while the NET-mediated activation of the inflammasome further amplifies the inflammatory response through a feed-forward loop [41].

Despite the implication of NETs in various pathologies, very little is known about their association with IBD, although several studies have proposed their role in the pathogenesis of IBD of both adults [45,46] and pediatric patients [47]. Gut proteome analysis in patients with ulcerative colitis showed a high expression of proteins that are associated with NETs [45]. Whether NETs formation is causative, or is rather a result of the inflammation, is still not completely known. Although ROS production is enhanced in CD, and neutrophils may be more prone to NET formation [48], their formation in CD is still controversially described. For example, a recently published small-scale study proved the enhanced presence of NETs in the intestinal tissues of pediatric CD and UC patients [47]. Lehmann et al. also reported upregulated proteins belonging to the main components of NETs in both diseases (UC and CD) compared to controls [49]. However, in other studies NETs have been observed and correlated with inflammation in UC, but not in CD, pointing to the stimulation of the innate immune system in the etiology of UC [45]. There are also studies suggesting that the presence of NETs in IBD does not have to be necessarily considered as a detrimental factor [50]. On the other hand, there are studies pointing to a detrimental role of NETs in IBD. Anti-neutrophil cytoplasmic autoantibodies (ANCAs) are biomarkers for the diagnosis and prognosis of IBD that are considered to activate, complement, and cause endothelial damage. They are known to target neutrophil proteins, which all are released during NET formation, suggesting this process might be the general cause of ANCAs production in IBD [51]. For example, ANCAs against myeloperoxidase were detected in the serum of many IBD patients [52], and against leukocyte proteinase 3 were also detected in IBD patients, but more frequently in UC than in CD [53]. The results of Dinallo et al. showed that NET-associated proteins were over-expressed in the inflamed colon of UC patients as compared to CD patients, suggesting a role for NETs in sustaining mucosal inflammation in UC. The same authors described also NETs’ production from the circulating neutrophils of UC patients in response to stimulation by TNF-α, with diminishing NETs formation following successful anti-TNF-α treatment [46,54]. Interesting findings came from He et al., who found that NETs facilitate pro-coagulant activity in patients with IBD as well as thromboembolic events that are known to exacerbate this disease [55]. In their extended study, they found that patients with active UC or CD had significantly increased levels of cfDNA, nucleosomes, and NETs formation compared to patients with inactive disease. They demonstrated that NETs represent a central component in the initiation and progression of colitis through mediating inflammation cell infiltration, driving cytokines release and thrombotic tendency.

In this context, the presence of NETs, their relative abundance, as well as their specific content seems to provide candidate biomarkers for the differential diagnosis of various types of IBD. Moreover, they may also represent the possible therapeutic targets of IBD and UC specifically [56], possibly through the specific inhibition of NET release, which was shown to be able to attenuate DSS-induced colitis in mice [46]. However, a detailed association of NET formation with different types and severities of IBD remains to be elucidated.

## 6. Cell-Free DNA in IBD

As was mentioned above, cfDNAs are present in various body fluids and biological materials. They are detectable under physiological conditions, but their presence during various diseases and physiological states is more interesting, at least for biomedical implications [57,58]. However, there are only a few publications regarding cfDNA in association with IBD.

It is known that cfDNA activates innate immunity through the activation of several DNA-sensing pathways, including toll-like receptor 9 (TLR9), stimulator of interferon protein (STING), and a protein called absent in melanoma 2 (AIM2) [59]. CfDNAs are able to bind to TLR9 and induce the cascade, leading to an inflammatory response, indicating that cfDNA could serve as a potential marker of inflammation. Experiments to study the role of these pathways in IBD have already been performed on knock-out mice that are deficient in these receptors. Some of them have paradoxically confirmed the protective role of these pathways in IBD [60,61]. This effect is probably mediated by intestinal microbiota, although detailed mechanisms are not yet known. Likewise, it was shown that the activation of TLR9 with the agonist administered prior to the induction of colitis induced anti-inflammatory effects [62]. On the other hand, a different study showed that the administration of oligodeoxynucleotides with CpG motifs (bacterial cfDNA) that activate TLR9 significantly exacerbates the course of DSS-induced colitis [63]. It is clear that bacterial ecDNA derived from intestinal microbiota is a heterogeneous group of various DNA molecules with potentially diverse roles in the pathogenesis of IBD.

Molnár et al. indicated that the intravenous administration of colitic cfDNA into healthy mice displays protective effects against DSS-induced colitis by altering the expression of several TLR9-related and inflammatory cytokine genes [64,65]. Thus, cfDNA found in the plasma of mice with DSS-induced colitis seems to have anti-inflammatory properties, but only in a preventive manner, as was shown by its transfer to healthy mice before the onset of the disease. It is known that the activation of TLRs in response to pathogen or damage-associated molecular patterns is associated with autophagy [66], and that autophagy contributes to NETosis [67]. This suggests a direct link between cfDNA and autophagy. In fact, TLR9-mediated autophagy might be the underlying phenomenon behind the protective effect of colitic cfDNA preconditioning [68]. However, the detailed mechanisms of action, as well as the origin of such cfDNA, is not fully understood.

In 2003, Rauh et al. showed for the first time that it is possible to detect cfDNA in the serum of UC patients, and that this cfDNA contained a microsatellite alteration previously identified in mucosa cells from UC patients [69]. A valuable study came from Koike et al., who detected a significantly higher amount of cfDNA in the circulation of mice with DSS-induced colitis compared to the control group. They also found a positive correlation between plasma cfDNA concentration and clinical severity of UC [70]. These findings are consistent with our recent study, which demonstrated a higher concentration of plasma cfDNA in mice with DSS-induced colitis group compared to the control. Moreover, the levels of plasma cfDNA negatively correlated with deoxyribonuclease (DNase) activity in the colon tissue [71]. These findings indicate that colon cells might represent one of the major sources of colitic plasma cfDNA, which further triggers downstream events that contribute to the pathogenesis. However, these findings are in discrepancy with a previous study in which no difference in cfDNA concentrations was found between DSS colitic mice and healthy controls [72]. On the other hand, recent results further demonstrated that the concentration of total cfDNA in the plasma of mice is increasing in parallel with the progression of the disease [73]. However, the increase of total plasma cfDNA levels did not correspond with an increase in plasma cf-ncDNA or cf-mtDNA. On the other hand, the total amount of cfDNA produced specifically by colon tissue was only increased in the early stages of the disease, and the increase corresponded to the increase of nuclear and mitochondrial cfDNA subtypes. These data suggest the crucial role of local colonic processes in triggering the inflammation. However, the later stage-associated increase in plasma cfDNA seems to be of a different than colonic origin.

Low amounts of human DNA released from the epithelial cells of the gastrointestinal wall can be detected in human fecal matter. However, in the state of inflammation or presence of infectious agents, when greater amounts of damaged and dead cells are exfoliated from the intestinal wall, the amount of DNA rises. Vincent et al. demonstrated that the excretion of large amounts of human DNA in feces is a general outcome of intestinal inflammation and is associated with the risk of *C. difficile* infection [74]. Casellas et al. showed that fecal and also gut lavage fluid DNA correlated with the clinical index and endoscopic score in patients with UC. Such fecal DNA excretion was significantly higher in patients with active disease, suggesting that it could be used as a non-invasive technique for the assessment of disease activity [75] and, because fecal DNA concentration increased in relapsed patients, also as an objective instrument to use in the follow-up of patients [76]. In addition, it was recently revealed that DNA methylation plays an important role in autoimmune-related chronic inflammatory diseases [77]. DNA methylation is an important epigenetic modification, which can silence genes, but also can lead to the increase of gene copy number and induce tumors. It is also known that IBD increases the risk of colorectal cancer, which is known as colitis-associated cancer [78]. The study of Lehmann-Werman et al. suggested that methylation patterns can be used to detect cfDNA derived from intestinal epithelial cells. Based on their results, intestinal DNA markers in healthy plasma and stool reflect the established route of clearance of intestinal DNA via the lumen of the gut. However, unlike Casellas et al., they detected only a minimal baseline intestinal cfDNA signal in IBD patients, which was indistinguishable from that of healthy individuals, compared to patients with advanced colorectal cancer, which had a strong intestinal signal [79]. On the other hand, Bai et al. detected a gradientally increased level of cfDNA in colitis and colon cancer mice, compared with control. They also studied the level of circulating DNA methylation, which decreased in colitis and colon cancer compared with control. Their results suggest that cfDNA and its methylation level can be considered new markers for colitis-to-cancer transformation [80].

## 7. Mitochondrial and Nuclear Cell-Free DNA in IBD

Mitochondria play an important role in inflammatory processes as they participate in metabolism and cell death signaling. Stress conditions can lead to damage to mitochondria and formation and the release of mitochondrial-derived vesicles (MDVs) containing mtDNA [81]. Such cf-mtDNA can act as a damage-associated molecular pattern (DAMP) that activates neutrophils through TLR9 because of the similarity with bacterial DNA [82,83]. Inflamed gut mucosa in IBD represents an enriched source of DAMPs; however, the role of cf-mtDNA in IBD is relatively unknown [84]. It has been proven that cf-mtDNA can activate various inflammatory responses via TLR9 receptors, including NLRP3 inflammasomes and neutrophils, as well as other downstream pro-inflammatory signaling proteins such as TNFα and NFκB [85,86]. Circulating cf-mtDNA was observed in a variety of inflammatory diseases and in patients with acute injury [83,87,88,89,90,91]. Boyapati et al. published the first report showing that cf-mtDNA is released during active IBD [92]. Increased levels of cf-mtDNA in UC and CD patients were detected, which significantly correlated with blood, clinical, and endoscopic markers and diseases activity. Inflammatory cells in lamina propria expressing TLR9 were also higher in active IBD patients. Apart from that, mitochondrial damage in inflamed UC mucosa and higher levels of fecal cf-mtDNA were observed. In parallel, these results were identified in a mouse model of DSS-induced colitis and recently, they confirmed these findings and demonstrated that deletion of the Tlr9 gene in mice results in the attenuation of acute DSS-induced colitis, suggesting cf-mtDNA-TLR9 signaling as an important and targetable pathway in IBD [92]. Another study showed that in mice with DSS-induced colitis, levels of plasma cf-ncDNA increased with the increased duration of colitis, and were directly proportional to the number of NETs [70].

Our group has recently tried to determine the dynamics of total cfDNA as well as cf-ncDNA and cf-mtDNA during an animal model of IBD. However, the concentration of circulating cf-mtDNA and cf-ncDNA in mice with colitis was not significantly different throughout the course of colitis compared to control mice [73]. On the other hand, cf-mtDNA released specifically from the colon tissue increased significantly during the early stages of the disease, indicating that colon-derived mt-DNA might be a significant factor that drives the local colonic inflammation, whereas other subtypes are primarily involved in the systemic inflammation.

## 8. Cell-Free DNA as a Therapeutic Target

DNases are enzymes cleaving DNA, which were proposed for the therapy of diseases with increased levels of cfDNA [93]. DNase I deficiency results in the difficulty of removing DNA from nuclear antigens, and consequently promotes susceptibility to autoimmune disorders [94]. DNase I deficiency was found in patients with systematic lupus erythematosus [95], and the administration of DNase I has been shown to be an efficient therapeutic agent in cystic fibrosis [96]. A reduced DNase I activity was also observed in patients with IBD [97]. Thus, in the view of the above-mentioned findings, the targeted digestion of cfDNA using DNase represents a potential novel therapeutic approach for IBD. DNase can directly cleave the circulating cfDNA, but can also break the structure of NETs, thereby reducing their pro-inflammatory properties [98,99]. The administration of DNase was shown to be effective in several immune-mediated experimental models, including sepsis [98] and hepatorenal injury [100]. In our recent study, intravenous DNase I injection was tested as potential therapy of DSS-induced colitis. Despite some improvement in the biochemical markers of colitis, the overall therapeutic effect was not proved, possibly due to the rapid half-life of the enzyme in circulation [72]. In the light of decreased DNase activity in the colon of mice with colitis [71], the topical colonic administration of DNase might be an interesting approach to further clarify the role of colon-derived cfDNA and test the rationale of cfDNA-targeted therapy.

NETs as higher structures may also represent possible therapeutic targets in IBD, and UC specifically [56], possibly through the specific inhibition of NET release, which was shown to be able to attenuate DSS-induced colitis in mice [46]. One of the principles to prevent the action of NETs is also disrupting their structure using DNase. Other approaches include preventing the oxidative burst of neutrophils that are necessary for the release of decondensed chromatin from neutrophils into the environment (e.g., by inhibitors of NADPH oxidase), or preventing the citrullination of histones (inhibitors of PAD4-peptidyl arginine deiminase type 4) [101].

At last, another means of possible cfDNA-based therapy relies on the administration of exosomes from healthy intestine, which were found to have immunosuppressive effects with a potential to prevent the development of IBD in a murine model of colitis. On the other hand, their inhibition can exacerbate murine IBD [30]. In light of recent findings proving the absence of cfDNA in exosomes, the possible therapeutic effect of exosomes might as well have been mediated by structures other than cfDNA [102]. In addition, exogenous exosome-like extracellular vesicles released by nematodes were also described to possess immunoregulatory molecules, proteins, and specific miRNAs, which were able to protect mice against chemically-induced colitis. Such specific proteins and miRNAs have great potential in the development of drugs to prevent chronic inflammatory diseases such as IBD [103].

## 9. Cell-Free miRNA in IBD

The studies focusing on the role of cfRNA mostly analyze the role of small non-coding RNAs, which are more stable than mRNA. Hence, the majority of publications regarding the role of cfRNA in IBD are related mainly to miRNA. MiRNAs are short (18 to 24 nucleotides in length), endogenous, non-coding single-stranded RNAs [104]. The sequences of miRNAs are evolutionary conserved across species, and at the post-transcriptional level, they regulate the expression of nearly one-third of the genes in the human genome and are involved in many biological processes (e.g., development, cell differentiation, proliferation, apoptosis), suggesting their important role in the pathogenesis of various diseases [105]. MiRNAs act primarily as post-transcriptional regulators via mRNA degradation and/or translational repression via the formation of miRNA-induced silencing complex (miRISC). MiRISC post-transcriptional control occurs through the inhibition of translation elongation, protein degradation, ribosome drop-off, or reducing the number of ribosomes on target mRNAs. However, miRNAs also have specific nuclear functions, including the miRNA-guided transcriptional control of gene expression. The mechanisms of miRNA-mediated transcriptional control of gene expression have not been completely elucidated [106].

The aberrant expression of miRNAs in IBD was, for the first time, reported in 2008 by Wu et al. Since then, many other studies were published that focused on the altered expression of miRNA in IBD, on the specific miRNAs and their association with target genes, or on single nucleotide polymorphism present in miRNAs implicated in the pathogenesis of IBD [107]. Among them, the most interesting studies (preliminary) are related to circulating miRNAs (here referred to as cf-miRNAs) identified in IBD patients, as they are considered promising biomarkers of disease severity, activity, or as a potential therapeutic target in IBD (for more information, see below). Cf-miRNAs are stable compared with mRNAs, and because they are packaged in exosomes or microvesicles, they are resistant to nuclease digestion. They were detected in serum and plasma samples, but also in urine or saliva [108,109].

In association with IBD, many studies have been published that detected either increased or decreased levels of specific cf-miRNA in patients with UC or CD compared with healthy controls. Some studies have focused on the relationships between altered miRNA expression in circulation and those at the diseased tissues, since sometimes, the altered miRNA expression in the diseased tissue is different from that observed in the peripheral areas where the cf-miRNA were quantified [110]. In 2011, Wu et al. showed for the first time that cf-miRNAs can be used to distinguish active CD and UC from healthy controls [111]. Since then, others have identified various dysregulated cf-miRNAs [112,113,114,115,116], and confirmed that there are several cf-miRNAs that are able to distinguish CD from UC or IBD from healthy controls (Table 1). Among them, some of these cf-miRNAs identified by Paraskevi et al. were consistent with those identified by Wu et al. and two cf-miRNAs that can differentiate CD from UC, identified by Netz et al., seem to have a biological basis for their differential expression based on the literature and acquired knowledge. An increase of specific cf-miRNAs was identified also in pediatric CD patients, which significantly decreased after six months of treatment, suggesting their role as non-invasive biomarkers of disease state [117]. However, Jensen et al. did not confirm findings that cf-miRNA can differentiate CD patients from healthy controls. They identified six downregulated and three upregulated cf-miRNAs in patients with CD compared with controls. However, in their validation cohort, only one from these, has-miR-16, was significantly downregulated in CD patients, with an inadequate discriminative power [118]. Interesting findings came from Polytarchou et al., who not only identified differentially expressed cf-miRNAs in patients with UC compared with controls, but also detected four cf-miRNAs correlating with disease activity, which were found to have higher sensitivity and specificity values than C-reactive protein (CRP) [114]. Similar results were found during analysis of patients with CD, where 10 cf-miRNAs were differentially expressed in patients compared with controls; two of these cf-miRNAs also correlated with CD disease activity and exhibited higher correlation values compared with CRP. Furthermore, distinct miRNA signatures between CD patients with ileal and colonic involvement were also revealed [115]. A distinct signature was also observed in mouse models and consequently validated using sera from UC patients. Based on obtained results, it seems that such a signature could be an ideal biomarker for IBD, since it can distinguish individuals at risk, predict the type of inflammation and disease status in patients, and evaluate the response to therapeutics [119]. These findings point to increasing evidence that cf-miRNAs play a key role in the pathogenesis of IBD, as many specific cf-miRNAs were identified to have different expression within UC compared with CD or within active phase in comparison to inactive disease, and some of these cf-miRNAs were shown to better reflect mucosal inflammation than CRP (Table 1) [120,121,122,123,124].

## 10. Cell-Free lncRNA in IBD

LncRNAs represent molecules longer than 200 nucleotides, and are localized mainly in the nucleus, but also in the cytoplasm and in extracellular fluids. Their role in gene regulation includes the control of the flux of genetic information, such as chromosome structure modulation, transcription, splicing, mRNA stability, mRNA availability, and post-translational modifications. In addition, they present interaction domains for DNA, mRNAs, miRNAs, and proteins. The mechanism of action of lncRNAs is given by their cellular and temporal specificity. LncRNA also contribute to the mRNA and protein content in the cell by regulating adjacent protein-coding genes expression [125].

LncRNAs can be categorized into sense, antisense, intronic, bidirectional, and intergenic lncRNAs [126]. Their expression is lower than those of protein-coding genes and depends on tissue and development-stage characteristics, pointing to their regulatory role [127,128]. Therefore, they have been studied in association with various diseases, such as cancer, cardiovascular, and neurological diseases. To date, hundreds of lncRNAs have been shown to be differentially expressed in IBD patients in comparison with healthy controls, and in active versus inactive disease state [129,130,131]. In many of these genes, various single nucleotide polymorphisms were identified and were found to be associated with transcription binding factors, expression quantitative trait loci, or DNase peaks [132]. There is a growing body of evidence that cf-lncRNAs play a key role in the pathogenesis of various diseases, and could be used as non-invasive biomarkers [133,134,135]. Recently, deregulated lncRNAs in the plasma samples of CD patients were observed [136], and Wang et al. identified significantly upregulated cf-lncRNAs (KIF9-AS1, LINC01272) and significantly downregulated cf-lncRNA DIO3OS in IBD patients compared with healthy controls (Table 2) [137]. Therefore, cf-lncRNAs could be potentially used as non-invasive biomarkers also in IBD; however, further investigation and studies are needed.

## 11. Clinical Relevance of cfNAs in IBD Care

Beyond cfDNAs as therapeutic targets, which was discussed in previous sections, another potential utility of cfNAs in clinical care can be found in differential diagnostics of clinically not typical or borderline cases of IBD, in monitoring the subclinical phases of disease onset in patients having a high risk of IBD, or those in remission, and also in monitoring the effectiveness of therapy used in IBD patients. All of these are basically connected to the continuous non-invasive monitoring of disease activity from blood, stool, or salivary samples of IBD patients, or those at risk. Although yet not extensively covered in the literature for IBD, cfNAs as biomarkers of therapy response were described in diseases such as metastatic colorectal cancer [138], non-small cell lung cancer [139], minimal residual disease [140], or in organ transplant monitoring [141]. However, a high potential for IBD activity tracking, and therefore also for therapy monitoring, can be anticipated from several of the above-described findings, which are mainly based on the total amount of cfNAs or on the specific dynamics between different types and origins of cfNAs identified. The main findings are as follows (for citations, see the relevant paragraphs of this review): (1) elevated levels of total cfNAs during active IBD; (2) specific balance between exosomes having pro-inflammatory and anti-inflammatory contents; (3) presence, relative abundance, and specific content of NETs discriminating various types of IBD; (4) presence and relative abundance of exogenous cfNAs as a reaction to the impaired barrier function of the intestinal epithelium; and (5) higher amounts of human DNA in feces as a general outcome of intestinal inflammation and epithelial cell damage when IBD is active. Another possibility of therapeutic response monitoring can be hypothesized also through the detection of anti-inflammatory content (immunoregulatory proteins and cfNAs) of hookworm-released exosomes [103] in the stool of IBD patients undergoing helminthotherapy.

Specific cfNAs-based application, having extremely high value in the clinical care of IBD patients, can be the non-invasive monitoring of colitis-to-cancer transformation. Such application was shown to be feasible by the monitoring of cfDNA and its methylation level [80].

## 12. Conclusions

As we reviewed here, different types of cfNAs can be released to different body fluids by several regulated or unregulated processes under specific physiological and pathophysiological conditions. When considering the endogenous cfNAs in blood, these include “passive” processes of cell death (necrosis, mechanical damage, etc.), “active” processes of cell death (apoptosis, NETosis), and “active” processes of cell–cell communications (exosomes, microvesicles). In the case of exogenous cfNAs in circulation (from microbiome, parasites, or ingested food), on the other hand, these processes can include “active” transport (hijacking the physiological transport machineries of the host organism, or through the immunological sampling of antigens by dendritic cells) and “passive” transport (because of inefficient epithelial barrier functions resulting from pathology or injury). Other specific processes can take place in body fluids that are different from blood, such as lymph, cerebrospinal fluid, saliva, urine, or stool. In each case, they represent signals from the processes taking place in the organism as well as signals to make reactions of the organism happen. Knowing the exact meaning of these signals can be exploited to actively step in the relevant processes or passively observe these processes. In IBD, for example, this takes place through inhibiting pro-inflammatory signal generation/transduction, strengthening anti-inflammatory signal generation/transduction, or monitoring disease onset and therapeutic efficiency. Therefore, it is evident that cfNAs play important roles in various diseases, including IBD. These specific molecules can be used as potential markers of disease state, and can even differentiate between the active and inactive phases of the disease. Their analysis is relatively easy and cheap, given their presence in body fluids. Therefore, CfNAs represent a significant tool of liquid medicine, which is reaching clinical care. It promises a non-invasive, low-risk diagnostic technique for patients. However, a number of various cfNAs types are known; yet regarding their role in IBD, only a limited number of publications are available. Nevertheless, these represent a significant basis for future research, as they demonstrated that cfNAs can be a valuable tool to stratify patients and distinguish them from healthy people. Further research is needed in order to identify the overall pool of cfNAs, but it is equally, if not more important to characterize these molecules in terms of their information value, cellular and subcellular origin, molecular pathways, and biological aspects in general. By achieving this, they could contribute to the elucidation of the pathogenesis of IBD and to the development of new therapeutic strategies.

## Figures and Tables

**Figure 1 ijms-20-03662-f001:**
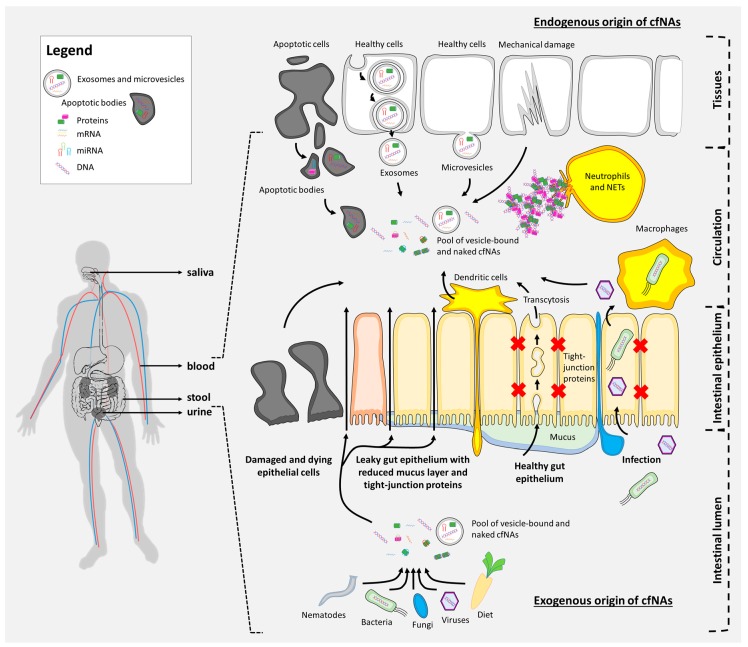
Origin, organization, routes of transfer, and availability for convenient sampling of cell-free nucleic acids (cfNAs). CfNAs can be readily isolated from various body fluids, including blood, saliva, urine, and stool, in which they are either naked or carried by various types of vesicles. Depicted are the most common sources of cfNAs in the circulation, which are relevant for inflammatory bowel disease (IBD), including endogenous sources (such as apoptotic bodies, exosomes, microvesicles, neutrophil extracellular traps (NETs), necrosis) as well as exogenous sources (such as diet and the intestinal microflora, including bacteria, fungi, nematodes, and viruses). Endogenous cfNAs most commonly originate from regulated or unregulated cell death (necrosis, mechanical damage vs. apoptosis, NETosis), or are secreted during processes of cell–cell communications (exosomes, microvesicles). Those of exogenous origin may reach the circulation by several processes such as hijacking the physiological transport machineries, transcytosis (vesicular uptake on one side of the epithelial barrier and release on the opposite side), continuous immunological sampling of antigens by dendritic cells, but also as a result of inefficient epithelial barrier functions during pathological processes. Common sources such as developing a fetus or tumors are not depicted, although the detection of colitis-to-cancer transition may have specific relevance for IBD patients. Apoptotic bodies may include also cellular organelles; however, these are not depicted. Since the figure focuses on the cfNAs content of the blood, content having endogenous origin in the intestinal lumen is not depicted, even though stool analysis may have specific relevance for the monitoring of health status and therapeutic effectiveness in IBD patients. Note that this depiction did not differentiate between single-layered and double-layered vesicles, and also that all the graphical components are illustrative and are not representative of real dimensions.

**Table 1 ijms-20-03662-t001:** Summary of identified nuclear (genomic) miRNAs (cf-miRNAs) deregulated in IBD. CD: Crohn’s disease, UC: ulcerative colitis.

cf-miRNA	Observed Change	Reference
miRs-199a-5p, miRs-362-3p, miRs-532-3p, miRplus-E1271	↑ in active CD patients	Wu et al., 2011 [111]
miRplus-F1065	↓ in active CD patients	Wu et al., 2011 [111]
miR-340*	↑ in CD patients	Wu et al., 2011 [111]
miR-149*	↓ in CD patients	Wu et al., 2011 [111]
miRs-28-5p, miRs-151-5p, miRs-199a-5p, miRs-340*, miRplus-E1271, miRs-3180-3p, miRplus-E1035, miRplus-F1159	↑ in active UC patients	Wu et al., 2011 [111]
miRs-103-2*, miRs-362-3p, miRs-532-3p,	↑ in UC patients	Wu et al., 2011 [111]
miR-505*	↓ in UC patients	Wu et al., 2011 [111]
miR-16, miR-23a, miR-29a, miR-106a, miR-107, miR-126, miR-191, miR-199a-5p, miR-200c, miR-362-3p, miR-532-3p	↑ in CD patients	Paraskevi et al., 2012 [113]
miR-16, miR-21, miR-28-5p, miR-151-5p, miR-155, miR-199a-5p	↑ in UC patients	Paraskevi et al., 2012 [113]
miRs-195, miR-16, miR-93, miR-140, miR-30e, miR-20a, miR-106a, miR-192, miR-21, miR-484, miR-let-7b	↑ in active CD patients	Zahm et al., 2011 [117]
miR-miRs-188-5p, miR-422a, miR-378, miR-500, miR-501-5p, miR-769-5p, miR-874	↑ in UC patients	Duttagupta et al., 2012 [112]
hsa-miR-369-3p, hsa-miR-376a, hsa-miR-376, hsa-miR-411#, hsa-miR-411, mmu-miR-379	↓ in CD patients	Jensen et al., 2015 [118]
hsa-miR-200c, hsa-miR-181-2 #, hsa-miR-125a-5p	↑ in CD patients	Jensen et al., 2015 [118]
miR-223a-3p, miR-23a-3p, miR-302-3p, miR-191-5p, miR-22-3p, miR-17-5p, miR-30e-5p, miR-148b-3p, miR-320e	↑ in UC patients	Polytarchou et al., 2015 [114]
miR-1827, miR-612, miR-188-5p	↓ in UC patients	Polytarchou et al., 2015 [114]
hsa-miR-1183, hsa-miR-1827, hsa-miR-1286, hsa-miR-504, hsa-miR-188-5p, hsa-miR-574-5p, hsa-miR-192-5p, hsa-miR-149-5p, and hsa-miR-378e	↓ in CD patients	Oikonomopoulos et al., 2016 [115]
hsa-miR-30e-5p	↑ in CD patients	Oikonomopoulos et al., 2016 [115]
miR-598, miR-642	↑ in UC patients	Netz et al., 2017 [116]
miR-595, miR-1246	↑ in active IBD	Krissansen et al., 2015 [120]
miR-223	↑ in IBD	Wang et al., 2016 [121]
miR-16, miR-21, miR-223	↑ in IBD, strongly in CD patients	Schonauen et al., 2017 [122]
miR-106a, miR-362-3p	↑ in IBD	Omidbakhsh et al., 2018 [123]
miR-146b-5p	↑ in IBD	Chen et al., 2019 [124]

**Table 2 ijms-20-03662-t002:** Summary of identified nuclear (genomic) long non-coding (cf-lncRNAs) deregulated in IBD.

cf-lncRNAs	Observed Change	Reference
KIF9-AS1, LINC01272	↑ in IBD	Wang et al 2018 [137]
DIO3OS	↓ in IBD	Wang et al 2018 [137]
GUSBP2, RP5-968D22.1, RP11-68L1.2, RP11-428F8.2, GAS5-AS1, RP11-923I11.5, DDX11-AS1, XLOC_005955, XLOC_005807, AC009133.20	↑ in CD	Chen et al., 2016 [136]
AF113016, ALOX12P2, AGSK1, CTC-338M12.3, AC064871.3, RP11-510H23.3, LOC729678, XLOC 010037, LOC283761, XLOC 013142	↓ in CD	Chen et al., 2016 [136]

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
