# Peer review of "Cell-Free Nucleic Acids and their Emerging Role in the Pathogenesis and Clinical Management of Inflammatory Bowel Disease"

_ijms, 2019, doi:10.3390/ijms20153662_

Round 1
Reviewer 1 Report
Manuscript IDijms-550871
Title: Type of the PaperCell-free nucleic acids and their 2 emerging role in the pathogenesis and clinical 3 management of inflammatory bowel disease
Dr. Kubiritova and colleagues provide a comprehensive review on potential roles of cfDNA and cfRNA in pathogenesis of IBD and UC.
Critique
Keywords should be limited to 10 words.
Line 52-56, to emphasize significance of NGS technologies in introduction is not appropriate. It should be moved to a different section
Lines 69-70: please consider revising the sentence.
Line 75: “Outbreak of cfNAs analyses....“ Please use different term (outbreak)
Lines 117-121, revise the paragraph or give an appropriate title to Figure 1, which shows “Origin and organization of cf-NAs."
Line 126: “4. Exosomes in IBD” This section describe exosomes and proteins associated with it. This section should focus on biological significance of nucleic acids that are associated with exosomes. Does nucleic acids associated with exosome considered cfNA? Needs a discussion.
Line 186-187 : “These traps are then able to capture the bacteria and neutralize them” The term neutralization should be explained or the changed.
Line 183: “Neutrophil extracellular traps in IBD” In previous paragraphs author state that cfDNAs are highly fragmented molecules to the size of approximately 160 bp, while mt DNA is fragmented to 30 to 80 bp. miRNAs are even shorter. Based on these data a short discussion is needed to explain how neutrophil extracellular trap complexes considered cfDNA.
Line 244: “……including TLR9, Sting and AIM2 [55]. CfDNAs are able to bind toll-like receptor 9 (TLR9). Please spell out TLR9 first. Same for Sting and AIM2.
Lines 345-347; “DNase can directly cleave the circulating cfDNA, but can also break the structure of NETs, thereby reducing their proinflammatory properties “ Please site paper or explain.
Line 409: Please be consistent microRNA or miRNA
Line 368: “Cell-free miRNA in IBD” A short paragraph is needed on how miRNAs regulate gene expression.
Line 403: delete “but”
Line 420 “Cell-free lncRNA in IBD“ A short paragraph is needed to describe roles of long non-coding RNAs in gene regulation
Conclusion: please summarize present understanding on role of cfNAs in IBD and UC instead of what is missing from the literature. “….. therefore a significant tool of liquid medicine which is reaching the clinical care …. “ “…… with regard to the role in IBD only a limited number of publications are available“ “…. these represent a significant basis for the future research….”
Author Response
Point-by-point response is uploaded.

Reviewer 2 Report
In the present review article, the Authors discussed the currently available data regarding the role of cell free nucleic acids in IBD. The manuscript is well written and provided a comprehensive description of the main findings regarding the involvemento of cfNA in the pathophysiology pf IBD. I have only few minor comments:
1) Maybe, a more detailed description of the origin of hexosomes could be helpful
2) At present, there is growing interest towerds the identification of specific markers of response to treatments in IBD. If available, it would be of interest to introduce comments from literature, or introduce a statement about the possibility to use cfNA as markers to treatment response.
Author Response
Point-by-point response is uploaded.
